# Adaptive Pseudo Text Augmentation for Noise-Robust Text-to-Image Person Re-Identification

**DOI:** 10.3390/s25237157

**Published:** 2025-11-24

**Authors:** Lian Xiong, Wangdong Li, Huaixin Chen, Yuxi Feng

**Affiliations:** 1School of Resources and Environment, University of Electronic Science and Technology of China, Chengdu 611731, China; xionglian@cqupt.edu.cn; 2School of Communications and Information Engineering, Chongqing University of Posts and Telecommunications, Chongqing 400065, China; lwd201723@gmail.com (W.L.); fyuxi957@foxmail.com (Y.F.)

**Keywords:** re-identification, cross-modal alignment, pseudo-text augmentation

## Abstract

**Highlights:**

**What are the main findings?**
The proposed Adaptive Pseudo Text Augmentation framework effectively identifies noisy image–text correspondence in text-to-image person re-identification using a Gaussian mixture model and recovers valuable data through pseudo-text generation with a multimodal large language model.The model achieves state-of-the-art or competitive performance on CUHK-PEDES, ICFG-PEDES, and RSTPReid datasets, demonstrating superior robustness under noisy supervision.

**What are the implications of the main findings?**
The proposed method improves the reliability of cross-modal alignment in real-world T2I-ReID datasets that contain coarse or mismatched text descriptions.It provides a general and extensible strategy for noise-resilient learning in other multimodal tasks involving imperfect annotations.

**Abstract:**

Text-to-image person re-identification (T2I-ReID) aims to retrieve pedestrians from images/videos based on textual descriptions. However, most methods implicitly assume that training image–text pairs are correctly aligned, while in practice, issues such as under-correlated and falsely correlated image–text pairs arise due to coarse-grained text annotations and erroneous textual descriptions. To address this problem, we propose a T2I-ReID method based on noise identification and pseudo-text generation. We first extracts image–text features using the Contrastive Language–Image Pre-Training model (CLIP), then employs the token fusion model to select and fuse informative local token features, resulting in token fusion embedding (TFE) for fine-grained representations. To identify noisy image–text pairs, we apply the two-component Gaussian mixture model (GMM) to fit the per-sample loss distributions computed by the predictions of basic feature embedding (BFE) and TFE. Finally, when the noise identification tends to stabilize, we employ a multimodal large language model (MLLM) to generate pseudo-texts that replace the noisy text, facilitating learning more reliable visual–semantic associations and cross-modal alignment under noisy conditions. Extensive experiments on the CUHK-PEDES, ICFG-PEDES, and RSTPReid datasets demonstrate the effectiveness of our proposed model and the good compatibility with other baselines.

## 1. Introduction

Text-to-image person re-identification (T2I-ReID) [1,2,3,4,5] aims to retrieve pedestrian images from large galleries based on natural language descriptions. This task plays a vital role in intelligent video surveillance and public security, where a witness or operator may only provide a textual description instead of an image. Despite the growing attention to this field, accurate cross-modal alignment between vision and language remains a challenging problem.

Most recent methods [6,7] assume that image–text correspondence in training datasets are correctly matched. However, in real-world datasets, textual descriptions are often incomplete, ambiguous, or even mismatched with their corresponding images. A noisy image–text correspondence refers to an image–text correspondence whose description is incomplete, inaccurate, or semantically inconsistent with the associated image (See Figure 1). It arises from coarse or erroneous manual annotations and inevitably degrades cross-modal learning. For example, in the RSTPReid dataset [8], many captions omit key visual attributes or describe irrelevant details, causing the model to learn incorrect associations between visual and textual modalities.

Existing T2I-ReID approaches primarily focus on designing better encoders or cross-modal alignment losses, while the issue of label noise has been largely overlooked. When trained on noisy data, these models tend to overfit spurious correlations and fail to learn robust semantic associations. Therefore, effectively identifying and correcting noisy image–text correspondence is crucial for achieving reliable retrieval performance.

To address this problem, we propose a model for text-to-image person re-identification under noisy environments using pseudo-text generation. Instead of discarding uncertain samples, our model first identifies potential noisy image–text correspondences through a probabilistic model and then regenerates reliable textual descriptions using an MLLM. The refined pseudo-texts preserve valuable training data while improving cross-modal consistency. Finally, we train the model jointly on both clean and pseudo-augmented pairs to achieve robust alignment under noisy supervision. The main contributions can be summarized as follows:
We present a novel noise-robust T2I-ReID framework that adaptively identifies and repairs noisy image–text correspondences instead of removing them.We design a Gaussian mixture model (GMM)-based noise identification mechanism combined with a pseudo-text generation module powered by a multimodal large language model, effectively recovering usable samples from noisy data.Extensive experiments on three public benchmark datasets, i.e., CUHK-PEDES [9], ICFG-PEDES [10], and RSTPReid [8], demonstrate the superiority of our method. Additionally, ablation experiments further confirm the effectiveness of each module in our method.

## 2. Related Work

### 2.1. Text-to-Image Person Re-Identification

The main challenge of T2I-ReID is to learn the mapping of visual and textual modalities into a common latent space. Early research [11,12,13,14,15,16] usually uses unimodal pre-trained models and attention mechanisms. For example, ResNet-50 [17] or ViT [18] is used as the image encoder and LSTM [19] or BERT [20] is used as the text encoder, and textual and visual features are fed into separate transformer blocks with self-attn and cross-attn independently to enable cross-modal interaction, with various loss functions designed to achieve cross-modal alignment. However, above methods initialize the network through single-modality pre-trained model parameters, ignoring the multi-modal correspondence information. In addition, the image and text embeddings extracted from the above-initialized network over-mine the information within a single modality, which increases the difficulty of cross-modal alignment and network optimization.

In recent years, vision-language pre-training (VLP) [21] has emerged as the prevailing paradigm in learning multimodal representations, demonstrating strong results on downstream tasks such as image captioning [22], visual question answering [23], and image–text retrieval [24]. Han et al. [25] pioneered the application of a CLIP [26] model for text-to-image person retrieval, leveraging a momentum contrastive learning framework to transfer knowledge acquired from large-scale generic image–text pairs. CFine [27] adopts the image encoder from CLIP to enhance cross-modal correspondence information, while replacing the text encoder with BERT to mitigate intra-modal information distortion. TP-TPS [6] explores the textual potential of CLIP in text-to-image person re-identification (T2I-ReID) by aligning images with constructed multi-integrity descriptions and attribute prompts. IRRA [7] incorporates an implicit relation reasoning module and introduces a masked language modeling (MLM) mechanism to align fine-grained information across modalities. RaSa [28] introduces two novel loss functions under the ALBEF [29] framework, specifically designed to improve performance in T2I-ReID tasks. Although these methods achieve promising performance, almost all of them implicitly assume that the training image–text pairs are correctly aligned, which is hard to meet in practice due to the ubiquitous noise.

### 2.2. Learning in Noisy Environments

Researchers have also paid attention to the noise problem and proposed relevant solutions. It can be mainly summarized into two approaches. (1) Sample selection [30,31,32]: This class of methods usually utilizes the memory effect of the deep neural networks (DNNSs) [33] to progressively differentiate and filter noisy data, but this may require additional computational resources and carry the risk of mistakenly deleting valid data. However, if the proportion of noisy samples is high then the model may not be able to learn effectively, resulting in overfitting the clean data, and the sample selection policy may cause selection bias, which affects the generalization ability of the model. (2) Robust loss functions [34,35,36]: This type of approach aims to develop noise-tolerance loss functions to improve the robustness of model training against noisy image–text correspondences. However, the computational overhead of this method increases dramatically when dealing with complex loss functions, affecting the training efficiency, and the loss function is so robust that the model may ignore the useful signals in the data, affecting the performance.

Recently, RDE [37] employs both sample selection and robust loss function to recognize and process noise, and achieves good results in noisy environments. However, for identified noisy image–text pairs, directly discarding them during network parameter updates means that only a subset of clean data is used for training in each epoch, leading to wasted data resources and limiting further improvements in model performance.

### 2.3. Multi-Modal Large Language Models

With the rapid advancement of multi-modal learning, multi-modal large language models (MLLMs) [38,39,40] have emerged as powerful tools for bridging vision and language modalities. These models integrate visual understanding and natural language generation capabilities, enabling automatic generation of high-quality textual descriptions for images. In the context of T2I-ReID, MLLMs offer a promising solution to address the problem of noisy image–text correspondence. Traditional manual annotations are prone to coarseness, inaccuracy, or semantic mismatch, while MLLMs can generate semantically aligned textual descriptions by directly understanding visual content. However, existing applications of MLLMs in T2I-ReID face key challenges, notably that simple static prompts often lead to homogeneous sentence patterns, causing the downstream T2I-ReID model to overfit to specific expression styles and reducing its generalization to real-world human descriptions [41].

To mitigate this limitation, Tan et al. [41] proposed a diverse text generation method based on dynamic template prompts, which extracts sentence patterns from existing dataset annotations to guide MLLM generation. This approach effectively improves the diversity of generated texts while maintaining consistency with real annotation styles. Building on this insight, recent works (e.g., Shao et al. [32]) have started to leverage MLLMs for pseudo-text generation to replace noisy annotations, but they often lack adaptive selection mechanisms to ensure semantic alignment between generated texts and images.

In our framework, different multimodal large language models play distinct roles. ChatGPT 4.0 is employed exclusively for template mining; by interacting with the model through multiple rounds of dialog, we extract and generalize common sentence structures from the original annotations. Qwen-VL-Chat-7B [42] is selected for pseudo-text generation due to its strong multimodal understanding capability, efficient local deployment, and stable performance on large-scale batch generation without relying on cloud APIs. This choice avoids potential privacy concerns and reduces cost, while maintaining high semantic alignment between generated descriptions and the associated images. We note that other models, such as MiniGPT-4 [40], BLIP [31], or LLaVA [38], may also be viable alternatives, each with trade-offs in generation quality, inference speed, and deployment requirements. To minimize hallucinations, dynamic prompts explicitly instruct the model not to invent any content absent from the image, and a randomly chosen template is integrated into each prompt to improve stylistic diversity.

## 3. Methodology

In this section, we present our proposed framework. To achieve robust fine-grained cross-modal alignment, particularly under noisy conditions, our framework employs three key components: feature representations for basic feature embedding (BFE) and token fusion embedding (TFE) features; adaptive pseudo-text augmentation, which first identifies noisy image–text pairs using a Gaussian mixture model (GMM) to focus subsequent processing on cleaner data, and then generates high-quality pseudo-texts for noisy images via am MLLM, effectively recovering valuable information from otherwise discarded samples; and cross-modal alignment, optimized using the triplet alignment loss (TAL), which ensures stable training by relaxing reliance on the hardest negative samples. These components work synergistically to enhance noise resilience. The overall framework of our method is illustrated in Figure 2: we detail each part next.

### 3.1. Feature Representations

In this section, we utilize the visual encoder and text encoder of the pre-trained CLIP to obtain token representations and implement cross-modal interactions through two token fusion modules.

Image encoder: The pre-trained ViT model of the CLIP is adopted to obtain the image embedding. Due to the mismatch between the image resolution of the dataset used for the text-to-image pedestrian re-identification task and the resolution of the original image on the WIT dataset, the positional embeddings in the pre-trained ViT model of the CLIP cannot be directly imported. In this paper, the linear 2D interpolation method used in TransReID [5] is employed to resize the positional embeddings for different image resolutions.

Given an input image $I\in R^{H\times W\times C}$, where H,W,C denote the height, width, and number of channels of the image, respectively, we divide the image into N non-overlapping image blocks of 16 × 16 pixels: $\left\{I_{i}|i=1,2,\ldots ,N\right\},N=\frac{H\times W}{P^{2}}$. The N image blocks are then projected through a linear projection layer to obtain N D-dimensional image block vectors, and a learnable [CLS] embedding vector is inserted in front of the input sequence of image block vectors. In order to learn the relative positional relationship between the image block vectors, a positional embedding $P\in R^{(N+1)\times D}$ is added to the input image block vector sequence, which can be expressed as follows:(1)$$X_{0}=\left[x_{\mathrm{cls}},E_{P_{1}}E_{P_{2}},\ldots ,E_{P_{N}}\right]+P,$$
where $p_{i}$ is a flattened patch, and **E** is a linear projection and denotes the linear projection layer that maps the parcel image to a D-dimensional vector. Then the embedding vector of image features is input into the ViT network. After passing through the L-layer transformer blocks the embedding vector $Z_{L}=ViT(X_{0}),f_{cls}^{v}=W_{v}Z_{cls}^{L}$ will be mapped to the joint image–text embedding space using linear projection, where $f_{cls}^{v}$ is used as the global feature representation of the image.

Text encoder: For the input text T, we directly use the CLIP text encoder to extract the textual representation, and following IRRA [7] we first tokenize input text T with a 49,152 vocab size into a token sequence using lower-cased byte pair encoding (BPE). The [SOS] and [EOS] embedding vectors are inserted at the beginning and the end of the text description to identify the beginning and the end of the text description sentence, respectively. The maximum length of the text description sequence is set to 77 in order to ensure the computational efficiency. In order to learn the relative positional relationship of the words in the sentence, the positional embedding $P\in R^{77\times D}$ is also added to the input sequence of word vectors. After the projection transformation, the input sequence can be represented as:(2)$$Y_{0}=\left[y_{cls},E_{t}w_{1},E_{t}w_{2},\ldots ,E_{t}w_{S},y_{eos}\right]+P_{t}$$

The final text-specific embedding vector input to the transformer can be expressed as:(3)$$\{f_{sos}^{t},f_{1}^{t},f_{2}^{t},\ldots ,f_{eos}^{t}\}$$

Similarly, the output text feature embedding vectors are projected into the joint image–text embedding space at the last layer of the transformer, and the vector $f_{eos}^{t}=W_{t}Z_{eos}^{L}$ represented by [EOS] is regarded as the global text feature representation. To compute the similarity between any image–text pairs (I, T), we directly utilize the global features [CLS] and [SOS] of the image and text to obtain the basic feature embedding (BFE) similarity by cosine similarity, i.e.,(4)$$S_{BFE}=\cos(f_{cls}^{v},f_{eos}^{t})=\frac{f_{cls}^{v}f_{eos}^{t}}{\left\|f_{cls}^{v}\|\|f_{eos}^{t}\right\|}$$

Token fusion: To more precisely capture the correspondence between modalities, we introduce local features for a finer-grained interaction between the two modalities. Since the global features ([CLS] and [EOS]) are obtained by weighted aggregation of all local features, and the correlation weights reflect the correlation between global features and individual local features, the more informative token can be utilized to learn more discriminative embedding representations to obtain more representative global features. According to the previous approach [37], we can select local features with higher scores for fusion from these correlation weights, which can be directly extracted from the self-attention weights of the last layer of the Transformer module. By selecting a proportion (σ) of the corresponding local token features based on correlation weights, we perform feature transformation to obtain more expressive representations. The feature transformation utilizes the same embedding module as in the residual block [17], which is as follows:(5)$$\left\{\begin{array}{c} f_{TFE}^{v}=\mathrm{MaxPool}(\mathrm{MLP}({\hat{f}}_{S}^{v})+\mathrm{FC}(f_{S}^{v}))\\ f_{TFE}^{t}=\mathrm{MaxPool}(\mathrm{MLP}({\hat{f}}_{S}^{t})+\mathrm{FC}(f_{S}^{t})) \end{array}\right.,$$
where $f_{S}^{v}=\left\{f_{1}^{v},f_{2}^{v}\cdots f_{k}^{v}\right\}$ denotes the top-K most informative local features selected, and ${\hat{f}}_{S}^{v}=L2Norm(f_{S}^{v})$ and ${\hat{f}}_{S}^{t}=L2Norm(f_{S}^{t})$ are the image and text features after L2-normalization. Finally, we calculate the cosine similarity $S_{TFE}$ between $f_{TFE}^{v}$ and $f_{TFE}^{t}$, and the global feature similarity embedding $S_{BFE}$, in order to evaluate the cross-modal matching degree during both training and inference.

### 3.2. Pseudo-Text Augmentations

#### 3.2.1. Noise Identification

To capture fine-grained correspondences between local visual patches and textual tokens, we propose token fusion embedding (TFE). TFE selects the top-K informative tokens from image and text token sequences according to the last-layer self-attention correlation scores, applies a residual embedding block to each selected token, and aggregates them via L2-normalized fusion. Deep neural networks tend to learn from clean samples faster than from noisy ones, because clean data generally results in lower loss values compared to noisy data [37,43]. Therefore, we utilize two two-component Gaussian mixture models, GMM to fit the per-sample loss computed by BFE and TFE to model the similarity of image–text pairs as clean distribution and noisy distribution, i.e., the Gaussian component with the lower mean value was assigned as the clean set and the other as the noisy set, respectively.

For the *i*th sample pair we define its TAL as $l_{i}$ (l is defined in Equation (10)). The per-sample loss is input into the GMM, which is optimized using the Expectation-Maximization (EM) [37] algorithm, then we compute the posterior probability, i.e.,(6)$$p(k|l_{i})=p(k)p(l_{i}|k)/p(l_{i})$$
where the posterior probability $p(k|l_{i}),k\in \{0,1\}$ denotes the probability that the *i*th sample is classified as a clean or noisy sample pair. By setting a threshold δ, thus we can classify the dataset containing M image–text pairs into a clean dataset ${\hat{S}}^{c}$ and noisy dataset ${\hat{S}}^{n}$. This can be depicted as follows:(7)$${\hat{S}}^{c}=\overset{M}{\underset{i=1}{\cup }}p\left(k=0\left|l_{i}\right.\right)>\delta ,{\hat{S}}^{n}=\overset{M}{\underset{i=1}{\cup }}p\left(k=1\left|l_{i}\right.\right)<\delta$$

We set the noisy dataset obtained from the two Gaussian mixture model GMMs as ${\hat{S}}_{gfe}^{n},{\hat{S}}_{tfe}^{n}$, respectively, and the final segmentation yields a noisy dataset of $S^{n}$ and a clean dataset of $S^{c}$. This can be depicted as follows:(8)$$S^{n}={\hat{S}}_{gfe}^{n}\cap {\hat{S}}_{tfe\;}^{n},S^{c}={\hat{S}}_{gfe}^{c}\cap {\hat{S}}_{tfe\;}^{c}$$

For the rest of the data, i.e., $S^{\prime }=S-\left(S^{c}\cup S^{n}\right)$, we randomly classify it as either a noisy set or a clean set. Instead of randomly assigning ambiguous samples, we adopt a soft-label weighting scheme based on their GMM posterior probabilities. Each sample *i* contributes to the loss as:(9)$$L_{i}=p_{i}\cdot L_{\mathrm{clean}}(i)+(1-p_{i})\cdot L_{\mathrm{noisy}}(i)$$
where $p_{i}$ is the posterior probability of being the clean samples. Samples with posterior values in the range [0.4, 0.6] are temporarily excluded from gradient updates, forming an uncertainty zone that prevents unstable supervision. This modification enhances training robustness and avoids reintroducing label noise into the optimization process.

#### 3.2.2. Pseudo-Text Generation

To replace manual annotations with high-quality textual descriptions, we utilize MLLM to generate textual descriptions automatically and employ them to replace traditional manual annotations. Specifically, we apply a similar method by providing the descriptions from the CUHK-PEDES, ICFG-PEDES, and RSTPReid datasets to GPT-3.5 [44] to capture their sentence patterns (i.e., description templates). After multiple rounds of dialog, ChatGPT generated 35 templates. We randomly select one of these templates and insert it into the static instruction to obtain the dynamic prompt as follows:

“Generate a description about the overall appearance of the person, including clothing, shoes, hairstyle, gender, and belongings, in a style similar to the template: ‘{template}’. If some requirements in the template are not visible, you can ignore them. Do not imagine any contents that are not in the image.”

The publicly available Qwen-VL [42] covers a broader range of capabilities, and its lightweight and convenient deployment makes it an ideal choice. Therefore, we select the Qwen-VL model for pseudo-text generation, as illustrated in Figure 3.

Once the model training process approaches a state of convergence, images derived from the noisy image–text pairs that were classified in the preceding training epoch are autonomously inputted into the MLLM. By leveraging dynamic prompting techniques, image captions are generated. These newly produced captions subsequently substitute the originally noisy text within the dataset. The model training then proceeds, utilizing this refined dataset for subsequent iterations of the training process.

Decoding and hyperparameters: We used the following decoding hyperparameters unless otherwise noted: temperature = 0.3, top_p = 0.95, max_length = 64 tokens, and num_return_sequences = 5 (i.e., generate 5 candidate captions per image). The random seed for generation was fixed at 42 to improve reproducibility.

Post-generation filtering and selection: To reduce hallucination and improve semantic alignment, we apply a two-stage selection procedure: (1) Heuristic filtering removes candidates that contain tokens that strongly contradict the image (e.g., explicit object labels known absent from dataset, or uses phrases like “not visible” that indicate model uncertainty). (2) CLIP reranking—for all remaining candidates we compute CLIP cosine similarity between the image and each candidate text; we select the caption with the highest CLIP similarity as the pseudo-text replacement. If all candidates are filtered, the top CLIP-scored candidate is selected (fallback). This selection strategy ensures the chosen pseudo-text both matches the image semantically and follows dataset-style templates.

When generation occurs we generate pseudo-texts after the noise-identification step stabilizes; in our experiments, stabilization of noise identification is observed around the 40th training epoch thus pseudo-text generation is executed at epoch 41. Generated pseudo-texts replace the noisy captions for subsequent training epochs.

#### 3.2.3. Cross-Modal Alignment

The commonly used loss functions in text-to-image matching systems that exhibit good performance include InfoNCE loss [45], Triplet Ranking loss (TRL) [46] and image–text similarity distribution matching (SDM) loss [7]. However, these loss functions face challenges such as handling the hardest negative samples and selecting appropriate similarity margins, particularly when dealing with the re-recognition of pedestrian text in noisy environments. Over-concern about the hardest negative samples may lead to local minima or even model collapse in such scenarios. In contrast, the novel triplet alignment loss (TAL) proposed by RDE [37] reduces the risk of the optimization being dominated by the hardest negatives.

By introducing a temperature coefficient τ, TAL considers all positive and negative sample pairs and relaxes the similarity learning from the hardest negative samples to all negative ones by applying an upper bound and using a weighted average to sum the log values of all negative samples, thereby making the training more stable and comprehensive by considering all pairs. Therefore, we utilize the TAL function to optimize and update the model. TAL is defined as:(10)$$l_{tal}(I_{i},T_{i})=m-{f_{v}^{i}}^{T}\overset{\wedge }{f_{t}^{i}}+\tau \log(\sum_{j\neq i}^{K}e^{\left({f_{v}^{i}}^{T}f_{t}^{i}/\tau \right)})$$
where *m* denotes the margin value of the difference between positive and negative samples, $f_{v}^{i}$ and $f_{t}^{i}$ denote the image and text of a batch of samples, respectively, and ${\hat{f}}_{t}^{i}$, $f_{t}^{j}$, and $f_{t}^{i}$ are the hardest negative, negative, and positive samples of the anchor sample $f_{v}^{i}$, respectively.

## 4. Experimentation

### 4.1. Datasets and Experimental Settings

#### 4.1.1. Dataset

We conduct experiments on three publicly available benchmark datasets using a common T2I-ReID evaluation metric to validate the effectiveness of the proposed algorithm.

CUHK-PEDES [9] contains 40,206 images and 80,412 text descriptions for 13,003 identities. Each image is manually annotated with two text descriptions, and the sentences contain rich details about the person’s appearance, movement, and pose. According to the official data split, the training set consists of 11,003 identities, 34,054 images, and 68,108 text descriptions.

ICFG-PEDES [10] includes 54,522 images from 4102 identities. Each image has a text description. The training set consists of 34,674 image–text pairs corresponding to 3102 identities, while the testing set contains 19,848 image–text pairs from the remaining 1000 identities. Second, the images contained in this dataset are more challenging due to their complex backgrounds and varied lighting changes, resulting in greater appearance variability.

RSTPReid [8] contains 20,505 images of 4101 identities from 15 cameras. Each person has five corresponding images taken by different cameras, and each image is annotated with two text descriptions. These images have complex indoor and outdoor scene transformations as well as backgrounds, making RSTPReid more challenging and better adapted to real scenarios.

#### 4.1.2. Evaluation Metrics

We employ the commonly adopted rank-k metrics (with k = 1, 5, and 10) as the main evaluation criteria. Rank-k measures the likelihood of identifying at least one correctly matched person image within the top-k ranked candidates when a textual description is used as the query. Additionally, for a more thorough evaluation, we also incorporate mean Average Precision (mAP) and mean Inverse Negative Penalty (mINP) [47] as alternative retrieval measures. Higher values of rank-k, mAP, and mINP indicate superior model performance.

#### 4.1.3. Experimental Details

As mentioned earlier, we adopt the pre-trained model CLIP as our modality-specific encoder. The input image resolution is 384 × 128, and the maximum length of the input text token is set to 77. During training, we introduce data augmentations to increase the diversity of the training data. Specifically, we apply random horizontal flipping, random cropping, and random erasing as data augmentation for the input images. Text augmentation involves random masking, replacement, and removal of word tokens. Our model is trained with an Adam optimizer for 60 epochs with a cosine learning rate decay strategy. The model training parameters follow RDE [37], where the initial learning rate is 1e − 5 for the original model parameters of CLIP and the initial one for the network parameters of TFE is initialized to 1e − 3. The batch size is 64. For the hyper-parameter settings, the margin value m of TAL is set to 0.1, the temperature parameter τ is set to 0.015, and the selection ratio σ is 0.3. The version of MLLM is Qwen-VL-Chat-7B, and all experiments were performed on a single NVIDIA GeForce RTX 4090D GPU with Pytorch 2.1.

### 4.2. Noisy Image–Text Pairs Identification

We perform the identification of image–text pairs on three public benchmark datasets to verify the effectiveness of the noise recognition module proposed in this paper.

From Figure 4, it can be seen that the noisy dataset identified by the noise identification module tends to stabilize when the training reaches around the 40th epoch. The underlying logic of effective noise identification is as follows: (1) During the early training stage (1st–20th epoch), the model’s loss calculation for clean and noisy samples is unstable, leading to fluctuations in the number of identified noisy samples (reflected by the fluctuating curves). (2) As training progresses (21st–40th epoch), the model gradually learns discriminative features, and the per-sample loss distributions of clean and noisy samples become separable. The two-component GMM effectively fits these two distinct distributions—clean samples cluster around the low-loss Gaussian component, while noisy samples cluster around the high-loss component—resulting in a gradual decrease in the fluctuation of the identified noisy sample count. After the 40th epoch, the noise identification results are stable and reliable; for example, RSTPReid (with the highest noise ratio due to coarse-grained annotations) stabilizes at approximately 9800 noisy samples, which accounts for ~47.8% of its total training samples—consistent with the statistical analysis of dataset noise in related works [8,37]. This stable identification result provides a solid foundation for the subsequent pseudo-text generation (Section 3.2.2), confirming that the GMM-based noise identification is effective in distinguishing noisy image–text correspondences from clean ones.

### 4.3. Comparison with State-of-the-Art Methods

In this section, we present comparison results with state-of-the-art methods on three public benchmark datasets. Based on the experimental results in Section 4.2, we set up the 40th epoch to identify noisy image–text pairs and generate pseudo-texts at the 41st epoch. Then, the generated image–text pairs are used to replace the noisy pairs for further model training. The experimental results are shown in the following Table 1, Table 2 and Table 3, and we annotate the feature extraction backbones (“Image Enc.” and “Text Enc.” column) employed by each method.

ICFG-PEDES: The experimental results on the ICFG-PEDES dataset are reported in Table 1, where the method proposed in this paper exceeds the previous state-of-the-art methods in Rank-1, Rank-5, Rank-10, and mAP, reaching 68.20%, 83.19%, 88.23%, and 41.36%, respectively. However, the mINP metric is relatively low at 7.89%, indicating that the inferior capability of our method to find the hardest matching samples. This limitation stems from the TALs’ balanced optimization across all positive–negative pairs, resulting in suboptimal optimization for challenging hard-negative samples.

RSTPReid: The experimental results on the RSTPReid dataset are shown in Table 2. Our proposed method achieves 66.24%, 85.89%, and 91.37% in Rank-1, Rank-5, and Rank-10, respectively, surpassing the recent state-of-the-art method RaSa in both Rank-1 and Rank-10 metrics. Also, due to the employment of TAL, the mINP metrics are not optimal.

CUHK-PEDES: We also report our experimental results on the CUHK-PEDES dataset in Table 3, where our method Rank-1 reaches 75.82%, slightly behind the best method, RaSa, but outperforms in mAP with a score of 67.58%. The relatively lower Rank-1 score on this dataset can be attributed to the high-quality annotations and detailed descriptions in the CUHK-PEDES dataset, which contain minimal noise. The noise identification module may occasionally misclassify certain clean image pairs as noisy, which slightly degrades the model’s performance on this dataset. In contrast, the ICFG-PEDES and RSTPReid datasets feature more challenging scenarios with higher levels of noise due to complex backgrounds and variable illuminations. In these cases, our method’s noise identification and pseudo-text generation capabilities enable superior performance.

### 4.4. Ablation Study

In this subsection, we analyze the effectiveness of each component in our framework, i.e., token fusion embedding, pseudo-text generation, and loss functions, on three publicly available benchmark datasets.

#### 4.4.1. Effectiveness of Token Fusion Embedding

To demonstrate the effectiveness of the token fusion, we conduct experiments by removing token fusion embedding (TFE), removing basic feature embedding (BFE), and combining both models. The results are presented in Table 4, Table 5 and Table 6, in which the combination of TFE and BFE achieves optimal performance across all metrics. This indicates that the combination provides a more comprehensive representation of both image and text, enabling the model to capture finer-grained interactions between the two modalities. Removing the TFE module leads to a noticeable drop in Rank-1 and mAP across all datasets (−3.8% and −2.7% on average, respectively). This indicates that TFE effectively captures fine-grained token-level correspondences, particularly for images containing multiple local attributes (e.g., accessories or complex backgrounds). Without token fusion, the model tends to focus only on global semantics, resulting in reduced discrimination for visually similar identities.

When using a combination of TFE and BFE, the performance increase is even more pronounced in noisy datasets. This confirms that our model mitigates the negative influence of incomplete or inaccurate annotations by enriching textual representations with auxiliary, semantically aligned descriptions. Notably, pseudo-text augmentation provides greater benefit to short or coarse captions, as it compensates for missing visual attributes that are often omitted in noisy text annotations.

#### 4.4.2. Effectiveness of Pseudo-Text Generation

To evaluate the impact of pseudo-text generation at different stages of training, we conduct experiments where pseudo-texts are generated at various epochs. Table 7, Table 8 and Table 9 report the results for ICFG-PEDES, RSTPReid, and CUHK-PEDES. “Generate 1”, “Generate 20”, and “Generate 40” denote the epoch at which pseudo-text generation is applied, while “Without” indicates that no pseudo-texts were used during training.

Across all datasets, pseudo-text generation consistently improves retrieval performance. Generating pseudo-texts at epoch 40 yields the largest gains, with improvements of +1.44% R-1 on ICFG-PEDES, +1.38% R-1 on RSTPReid, and +1.86% R-1 on CUHK-PEDES compared with the baseline without pseudo-texts. These improvements align with our observation that the GMM-based noise estimation stabilizes around epoch 40, allowing pseudo-text generation to replace noisy annotations with high-quality descriptions at the most appropriate time. The results demonstrate two important properties of the pseudo-text mechanism:

Consistency. Even early pseudo-text replacement (e.g., epoch 1 or 20) does not degrade performance and generally improves it.

Noise-awareness. The largest improvement occurs when pseudo-text replacement leverages stable noise estimates, demonstrating the benefit of deferring generation until the noise-identification step becomes reliable.

#### 4.4.3. Sensitivity Analysis of the Model

To further interpret the effectiveness of our approach, we conducted multi-seed and sensitivity analyses. The multi-seed results show stable performance variations (±0.3% R-1 and ±0.4% mAP), confirming that our model maintains consistent alignment accuracy under different random initializations.

We additionally examine how the batch size and loss-related hyperparameters affect model behavior. On the RSTPReid dataset, we systematically varied the margin *m* and temperature *τ* in the RSTPReid dataset under identical conditions. The results (Appendix A) show that performance peaks at *m* = 0.1: both overly small (<0.05) and large (>0.15) margins reduce retrieval accuracy. Similarly, the temperature parameter is critical for stability: values below *τ* = 0.01 cause instability, while *τ* > 0.02 degrades discrimination. Based on this, we selected *m* = 0.1 and *τ* =.015 as optimal. These observations confirm the robustness and generalizability of the proposed pseudo-text enhancement mechanism across different noise levels.

## 5. Conclusions

The quality of training data, particularly the issues of noisy image–text correspondence, is a critical concern in text-to-image person re-identification (T2I-ReID). These issues can lead models to learn erroneous image–text associations, ultimately resulting in overfitting. In this paper, we investigate the noisy image–text correspondence in T2I-ReID, which challenges the common assumption in existing methods that image–text data is perfectly aligned. Therefore, we propose a T2I-ReID method based on noise recognition and adaptive pseudo-text generation. The method leverages CLIP for extracting image–text features and employs the Gaussian mixture model (GMM) for identifying noisy image–text pairs. Subsequently, we utilize ChatGPT to capture textual description templates from the dataset, generating dynamic prompts in the process. These dynamic prompts, along with the identified noisy images, are fed into a multimodal large language model to produce diverse and accurate pseudo-text descriptions. The generated pseudo-texts are subsequently used to replace the noisy texts, forming clean image–text pairs that are utilized for model training with the TAL function to achieve cross-modal alignment. Experimental results demonstrate that our proposed method achieves competitive or state-of-the-art performance on the CUHK-PEDES, ICFG-PEDES, and RSTPReid benchmark datasets for text-to-image person re-identification, particularly in mitigating the effects of label noise.

Although the proposed framework demonstrates strong robustness against noisy annotations, its effectiveness still depends on the pseudo-text generation quality. In cases where the multimodal language model produces incomplete or contextually inconsistent descriptions, slight performance degradation may occur. Future work will explore integrating feedback-based refinement mechanisms and contrastive self-correction strategies to further enhance text reliability. Additionally, we plan to extend our method to open-domain vision–language datasets beyond person re-identification.

## Figures and Tables

**Figure 1 sensors-25-07157-f001:**
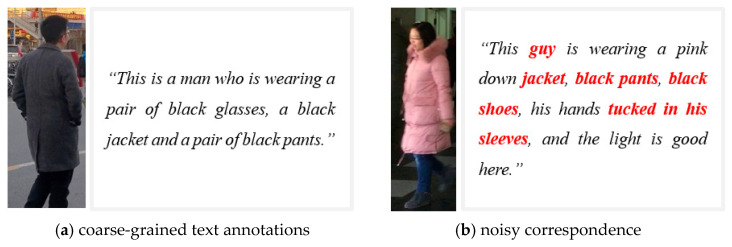
The illustration of noisy image–text correspondence. text granularity roughness. Note that both examples in (**a**,**b**) are from and actually exist in the RSTPReid dataset.

**Figure 2 sensors-25-07157-f002:**
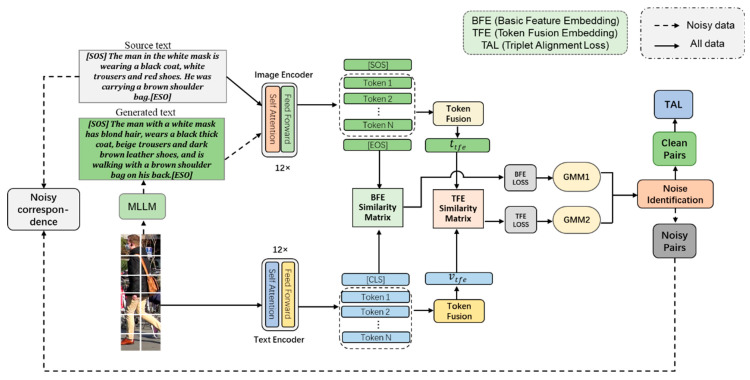
Framework of our method.

**Figure 3 sensors-25-07157-f003:**
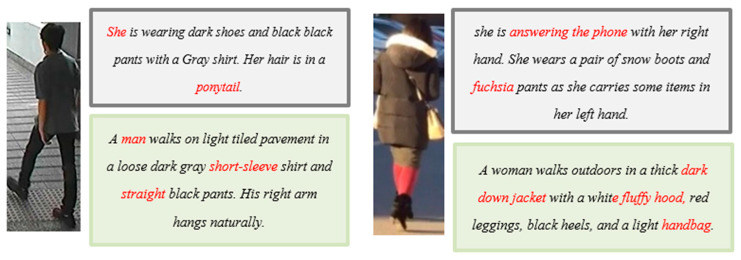
Schematic diagram of the pseudo-text generation. Where the upper gray background text is the original text of the dataset, the text at the bottom with a green background corresponds to the pseudo-text generated by the MLLM.

**Figure 4 sensors-25-07157-f004:**
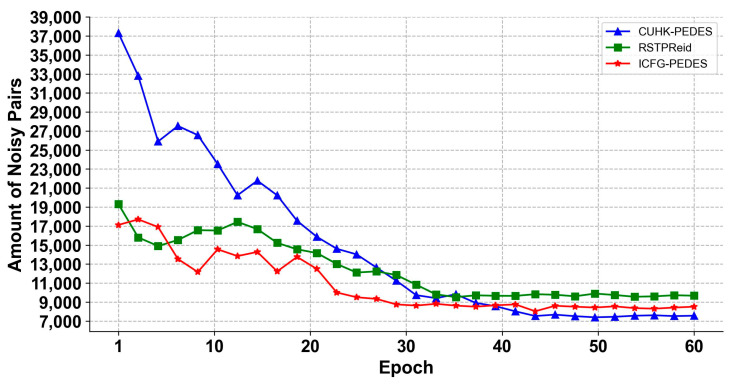
Amount of noisy image–text pairs identified in the three datasets.

**Table 1 sensors-25-07157-t001:** Performance comparisons with state-of-the-art methods on the ICFG-PEDES dataset. The bold font denotes the best performance.

Methods	Image Enc.	Text Enc.	ICFG-PEDES				
			R-1	R-5	R-10	mAP	mINP
SSAN [10]	RN50	LSTM	54.23	72.63	79.53	-	-
IVT [48]	ViT-Base	BERT	56.04	73.60	80.22	-	-
CFine [27]	CLIP-ViT	BERT	60.83	76.55	82.42	-	-
TP-TPS [6]	CLIP-ViT	CLIP-Xformer	60.64	75.97	81.76	41.27	-
IRRA [7]	CLIP-ViT	CLIP-Xformer	63.46	80.25	85.82	38.06	7.93
RaSa [28]	CLIP-ViT	BERT-base	65.28	80.40	85.12	41.29	**9.97**
RDE [37]	CLIP-ViT	CLIP-Xformer	67.68	82.47	87.36	40.06	7.87
Ours	CLIP-ViT	CLIP-Xformer	** 68.20 **	** 83.19 **	** 88.23 **	** 41.36 **	7.89

**Table 4 sensors-25-07157-t004:** Ablations of token fusion on the ICFG-PEDES dataset. The bolds are the best performance values.

Methods	ICFG-PEDES				
	R-1	R-5	R-10	mAP	mINP
Baseline with BFE	65.33	82.12	86.82	38.83	7.65
Baseline with TFE	67.72	82.96	87.45	40.93	7.83
Baseline with BFE + TFE	**68.20**	**83.19**	**88.23**	**41.31**	**7.89**

**Table 7 sensors-25-07157-t007:** Ablations of pseudo-text generation at different epochs on ICFG-PEDES. The bolds are the best performance values.

Methods	ICFG-PEDES				
	R-1	R-5	R-10	mAP	mINP
Without	66.76	82.65	88.63	40.53	7.57
Generate 1	66.87	82.34	87.43	40.32	7.68
Generate 20	67.12	82.56	88.41	41.02	8.32
Generate 40	**68.20**	**83.29**	**89.13**	**41.38**	**8.67**

**Table 2 sensors-25-07157-t002:** Performance comparisons with state-of-the-art methods on the RSTPReid dataset. The bolds are the best performance values.

Methods	Image Enc.	Text Enc.	RSTPReid				
			R-1	R-5	R-10	mAP	mINP
SSAN 10]	RN50	LSTM	43.50	67.80	77.15	-	-
IVT [48]	ViT-Base	BERT	46.70	70.00	78.80	-	-
CFine [27]	CLIP-ViT	BERT	50.55	72.50	81.60	-	-
TP-TPS [6]	CLIP-ViT	CLIP-Xformer	50.65	72.45	81.20	43.11	-
IRRA [7]	CLIP-ViT	CLIP-Xformer	60.20	81.30	88.20	47.17	25.82
RaSa [28]	CLIP-ViT	BERT-base	65.90	**86.50**	91.35	52.31	**29.23**
RDE [37]	CLIP-ViT	CLIP-Xformer	65.35	83.95	89.90	50.83	29.02
Ours	CLIP-ViT	CLIP-Xformer	**66.24**	85.89	**91.37**	**52.35**	29.09

**Table 3 sensors-25-07157-t003:** Performance comparisons with state-of-the-art methods on the CUHK-PEDES dataset. The bolds are the best performance values.

Methods	Image Enc.	Text Enc.	CUHK-PEDES				
			R-1	R-5	R-10	mAP	mINP
SSAN [10]	RN50	LSTM	61.37	80.15	86.73	-	-
IVT [48]	ViT-Base	BERT	65.59	83.11	89.21	-	-
CFine [27]	CLIP-ViT	BERT	69.57	85.93	91.15	-	-
TP-TPS [6]	CLIP-ViT	CLIP-Xformer	70.16	86.10	90.98	66.32	-
IRRA [7]	CLIP-ViT	CLIP-Xformer	73.38	89.93	93.71	66.13	50.24
RaSa [28]	CLIP-ViT	BERT-base	**76.51**	**90.29**	**94.25**	67.49	51.11
RDE [37]	CLIP-ViT	CLIP-Xformer	75.94	90.21	94.12	67.56	**51.44**
Ours	CLIP-ViT	CLIP-Xformer	75.82	90.26	94.23	**67.58**	51.27

**Table 5 sensors-25-07157-t005:** Ablations of token fusion on the RSTPReid dataset. The bolds are the best performance values.

Methods	RSTPReid				
	R-1	R-5	R-10	mAP	mINP
Baseline with BFE	63.45	85.18	90.26	50.78	28.02
Baseline with TFE	65.32	86.14	91.24	50.97	27.83
Baseline with BFE + TFE	**66.24**	**85.89**	**91.37**	**52.35**	**29.09**

**Table 6 sensors-25-07157-t006:** Ablations of token fusion on the CUHK-PEDES dataset. The bolds are the best performance values.

Methods	CUHK-PEDES				
	R-1	R-5	R-10	mAP	mINP
Baseline with BFE	72.89	88.67	92.45	65.92	49.89
Baseline with TFE	74.32	89.02	92.78	66.24	50.14
Baseline with BFE + TFE	**75.82**	**90.26**	**94.23**	**67.58**	**51.27**

**Table 8 sensors-25-07157-t008:** Ablations of pseudo-text generation at different epochs on RSTPReid. The bolds are the best performance values.

Methods	RSTPReid				
	R-1	R-5	R-10	mAP	mINP
Without	64.86	84.85	88.78	50.81	27.56
Generate 1	65.32	84.89	88.45	50.43	28.01
Generate 20	65.67	85.34	88.39	51.32	28.32
Generate 40	**66.24**	**85.89**	**91.37**	**52.35**	**29.09**

**Table 9 sensors-25-07157-t009:** Ablations of pseudo-text generation at different epochs on CUHK-PEDES. The bolds are the best performance values.

Methods	CUHK-PEDES				
	R-1	R-5	R-10	mAP	mINP
Without	73.96	89.89	93.08	67.19	51.13
Generate 1	73.07	89.73	93.34	67.22	50.96
Generate 20	75.19	90.21	94.16	67.39	**51.32**
Generate 40	**75.82**	**90.26**	**94.23**	**67.58**	51.27

## Data Availability

The datasets analyzed during this study are available in the following public domain resources. CUHK-PEDES dataset: https://github.com/layumi/Image-Text-Embedding/tree/master/dataset/CUHK-PEDES-prepare, accessed on 24 November 2017. RSTPReid dataset: https://github.com/NjtechCVLab/RSTPReid-Dataset, accessed on 12 September 2021. Requests for additional information should be directed to huaixinchen@uestc.edu.cn.

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
