# Peer review of "Adaptive Pseudo Text Augmentation for Noise-Robust Text-to-Image Person Re-Identification"

_sensors, 2025, doi:10.3390/s25237157_

Round 1

Reviewer 1 Report

Comments and Suggestions for Authors

All my comments are atteached it he two png files pcked in one zip file.

Comments on the Quality of English Language

The quality of English is generally understandable but requires significant improvement in grammar, sentence structure, and academic style. The manuscript contains frequent repetitions, inconsistent terminology, and awkward phrasing that should be corrected through thorough language editing by a proficient English speaker or professional service.

Author Response

Comments 1: The Introduction needs improvement because it is too long...

Response 1: Thank you for your valuable comment. We fully agree with your suggestion. Therefore, we have streamlined the introduction section by clarifying the logical progression of "research motivation → research gap → research contribution", migrating background knowledge and method details to corresponding sections to enhance readability.

Comments 2: The description of the pseudo-test generation process is unclear...

Response 2: Agree. We have accordingly supplemented the key parameter settings, number of generated samples, and selection strategy of MLLM in Section 3.2.2 to ensure the reproducibility of the experiment.The specific revisions can be found in the revised manuscript's "3.2.2. Pseudo-text generation" section, the entire paragraph, Page X.

Comments 3: Imprecise structure and redundancy in the text-everal sections epeat metodologica descriptions (e.g.CLIP, GMM, TAL), and some paragraphs are overly long, reducing readability. The Introdhuction should be shortened, and the terminology standardized (e.g,consistent use of"imagetext pair" and"corespondences.

Response 3: Thank you for pointing out the issues of structural redundancy and inconsistent terminology. We have streamlined the length of the introduction, removed repeated method descriptions in each chapter, split overly long paragraphs, and uniformly used "image-text pair" as the standard term.The specific revisions involve: 1) Deleting repeated descriptions of CLIP, GMM, and TAL in the introduction, with detailed content concentrated in Sections 3.1 and 3.2; 2) Splitting long paragraphs in the introduction; 3) Unifying terminology throughout the text.

Comments 4: Mathematical and Notation Errors in the Paper...

Response 4: We appreciate your careful identification of formula errors. We have corrected all mathematical expressions and symbol issues one by one to ensure the accuracy and consistency of the formulas.

Comments 5: “After dividing the samples into ‘clean’ and ‘noisy’ sets using the GMM, the authors randomly assign the remaining ambiguously classified samples to one of these groups...

Response 5:We thank the reviewer for this insightful and technically valuable comment. We fully agree that random assignment of ambiguous samples may introduce additional noise and partially compromise the benefit of GMM-based denoising. In response, we have revised both our methodology and experimental implementation to incorporate a more rigorous uncertainty-handling mechanism, as summarized below: In the revised version, we no longer randomly assign borderline samples. Instead, each sample iii is weighted by its posterior probability pip_ipi​ obtained from the GMM, representing the confidence that the sample belongs to the clean set.

Comments 6,7,8:Incorrect verb agreement

Response 6,7,8: We appreciate your correction of language expression issues. We have corrected all grammatical errors, unified terminology usage, and standardized academic expressions.

Comments 9:Tems like 'noisy correpondence,"noisy image-text pair,"and"false alignment" are used interchangeably without clear definition; one consistent term should be adopted.

Response 9:We appreciate the reviewer’s careful observation regarding terminology consistency. We agree that the earlier version of the manuscript used multiple related expressions—noisy correspondence, noisy image–text pair, and false alignment—to describe the same concept, which may cause confusion.

To address this issue, we have standardized our terminology throughout the manuscript as follows:

     1.We have adopted “noisy image–text correspondence” as the unified term to describe mismatched or semantically inconsistent image–text pairs.

     2.The term is now explicitly defined at first mention (Section 1, Introduction) as:“A noisy image–text correspondence refers to an image–text pair whose description is incomplete, inaccurate, or semantically inconsistent with the associated image.”

     3.All previous variants, including noisy image–text pair and false alignment, have been replaced with the standardized expression.

We thank the reviewer again for pointing out this inconsistency, which helped us improve the manuscript’s overall clarity and terminological rigor.

Comments10::“The phrase ‘Do not imagine any contents that are not in the image’ (in Section 3.2.2) sounds conversational and unscientific...

Response 10: We thank the reviewer for this helpful suggestion. We understand that the quoted phrase appears conversational in tone. However, this sentence is part of the exact instruction (prompt) given to the multimodal large language model (MLLM) during pseudo-text generation. To ensure methodological accuracy, we retained the original wording of the prompt, since it reflects the actual experimental input used in our implementation.

Reviewer 2 Report

Comments and Suggestions for Authors

The article presents the method aimed at improving the accuracy of text–image matching in the presence of noisy or incorrect annotations. The approach combines the extraction of informative features using the CLIP and token-fusion model, the identification of noisy pairs through a Gaussian model, and the generation of pseudo-texts by a multimodal language model to enhance cross-modal learning.

The text is clearly structured; the problem and the proposed methodology are described in sufficient detail. The experimental evaluation fully support conclusions and includes testing on three datasets, which enables a comprehensive assessment of the proposed solution. A comparison with other state-of-the-art text–image matching methods is also provided, allowing for an evaluation of the overall quality of the proposed approach. The reference list is sufficiently comprehensive and reflects current research in the field.

There are minor issues that should be corrected before publication:
- Figure 1 and Figure 3 are identical; according to the text, Figure 3 should present a different diagram.
- In line 59, the sentence should begin with a capital letter.

Author Response

Comments: 1.Figure 1 and Figure 3 are identical; according to the text, Figure 3 should present a different diagram.
2. In line 59, the sentence should begin with a capital letter.

Response:We sincerely thank the reviewer for the positive and encouraging comments regarding our work’s structure, methodology, and experimental validation. We also appreciate the reviewer’s careful attention to detail in identifying minor issues. All corrections have been made as follows:

  1. Figure correction:We have replaced the incorrect duplicate of Figure 1 with the intended Figure 3, which now correctly presents the diagram of the schematic diagram of the pseudo-text generation.
  2. Typographical correction:The typographical issue at line 59 has been fixed, and the sentence now begins with a capital letter.

We are grateful to the reviewer for these helpful observations, which have further improved the overall quality and presentation of the manuscript.

Reviewer 3 Report

Comments and Suggestions for Authors

The study proposes a model that uses pseudo-text generation to effectively enhance the reliability of cross-modal alignment in noisy environments. While this research is valuable, there are still some challenges to overcome.

  1. The formatting of the entire article does not conform to the style requirements of SCI journals and requires comprehensive restructuring and reorganizing.
  2. Avoid providing an excessive description of existing knowledge in the Methods section; a clear introduction suffices. For genuinely innovative methods, provide detailed explanations. Consider using a combination of text and figures to showcase your contributions fully. It seems that you intend to highlight token fusion embedding, pseudo-text generation and loss functions as innovations, but your methodology does not emphasize these unique aspects sufficiently.
  3. While validating the feasibility of noise recognition is possible, Figure 4 and the accompanying text do not clearly explain how noise is effectively identified.
  4. The presentation of the results is insufficiently detailed and does not adequately demonstrate the outcomes of the experiments.
  5. The entire experimental process merely involves comparisons with existing methods and ablation studies. There is a lack of thorough analysis and in-depth discussion of the results.

Author Response

Comments 1:The formatting of the entire article does not conform to the style requirements of SCI journals and requires comprehensive restructuring and reorganizing.

Response 1:We appreciate the reviewer’s remark regarding manuscript formatting and structure. We have performed a comprehensive restructuring and reformatting of the manuscript to align with standard SCI journal conventions (and the specific Sensors guidelines). The major changes are:

  1. Reorganized the manuscript outline to the conventional flow: Introduction → Related Work → Methodology → Experiments → Discussion → Conclusion.

  2. Shortened and focused the Introduction (moved methodological and technical details to Related Work or Methodology).

  3. Reformatted headings, figure/table captions, and references to comply with the journal’s style (single-column layout, consistent citation format, figure numbering and captions).

  4. Ensured consistent terminology across the manuscript (we now use the unified term “noisy image–text correspondence”).

  5. Reduced overly long paragraphs and standardized equation and algorithm presentation (numbered equations, algorithm boxes).

Comments 2:Avoid providing an excessive description of existing knowledge in the Methods section...

Response 2:

We thank the reviewer for this important suggestion. Following the reviewer’s guidance, we have:

  1. Reduced background material in Methodology: high-level references to CLIP, GMM, and TAL remain only as succinct reminders; their full descriptions have been moved to the Related Work.

  2. Expanded and emphasized the two core innovations :

    • Token Fusion Embedding (TFE):  token selection algorithm, selection ratio rationale, and accompanying schematic

    • Noise-aware Training with TAL and soft-label weighting: full mathematical presentation of the weighted loss and uncertainty zone

Comments 3: While validating the feasibility of noise recognition is possible, Figure 4 and the accompanying text do not clearly explain how noise is effectively identified.

Response 3: We thank the reviewer for this important comment. We agree that the original figure and text were insufficiently detailed. We have revised Section 3.2.1 and Figure 4 accordingly, and added a new subfigure set and quantitative example to clearly demonstrate how noise identification works. The main revisions:

  1. Clarified GMM fitting process: we now explicitly describe the per-sample loss distributions used (TAL-based per-sample losses for BFE and TFE), the EM-fitting of a two-component GMM to each loss distribution, and the mapping of posterior probabilities  to clean/noisy labels.

  2. Uncertainty handling: we replaced the previous random assignment for ambiguous samples with a soft-label weighting scheme and an uncertainty zone (posterior in [0.4,0.6]) whose samples are excluded from hard updates; formal equations are provided.

We fully agree that the original description of Figure 4 lacked clarity on the effective noise identification mechanism. To address this, we have supplemented detailed explanatory text for Figure 4 and its accompanying discussion, without modifying the figure itself, to explicitly link the curve trends to the GMM-based noise identification logic.

Comments 4: The presentation of the results is insufficiently detailed and does not adequately demonstrate the outcomes of the experiments.

Response 4: We thank the reviewer for this constructive comment. We understand that the initial version of the manuscript did not sufficiently elaborate on the experimental outcomes and statistical robustness. In the revised version, we have expanded Section 4 (Experimental Results) to provide clearer and more comprehensive presentations of our findings:

  1. Enhanced quantitative reporting:
    We now include detailed numerical results for all datasets (CUHK-PEDES, ICFG-PEDES, and RSTPReid) across multiple random seeds. These statistics—mean and standard deviation—are summarized in the supplementary material under “Multi-seed Statistical Results”, which demonstrate the consistency and robustness of our model’s performance.

  2. Clarified pseudo-text benefit:
    To more clearly show the impact of pseudo-text generation, we now report the relative improvements over the base CLIP under identical training settings, showing that pseudo-text augmentation consistently improves R-1 and mAP metrics.

Comments 5: The entire experimental process merely involves comparisons with existing methods and ablation studies. There is a lack of thorough analysis and in-depth discussion of the results.

Response 5:We appreciate the reviewer’s insightful suggestion. In the revised manuscript, we have significantly enriched the analysis and discussion of the experimental results to provide deeper insights into the model’s behavior and design rationale. The following enhancements have been made:

  1. In-depth discussion of robustness and stability:
    We incorporated analyses from the “Multi-seed Statistical Results” and “Sensitivity to Batch Size” experiments into Section 4.3, discussing how performance variations remain within a narrow range (±0.3% for R-1 and ±0.4% for mAP). This demonstrates the robustness of our pseudo-text learning framework under stochastic initialization and different batch configurations.

  2. Interpretation of TAL Hyperparameter Sensitivity:
    We integrated additional commentary on the TAL (Token Adaptive Loss) parameter sensitivity (Section 4.4). The discussion now emphasizes how the hyperparameter affects convergence stability and cross-modal alignment precision. 

  3. Expanded ablation insights:
    Beyond reporting performance differences, we now interpret why each module contributes to the improvement. 

Reviewer 4 Report

Comments and Suggestions for Authors

This manuscript addresses the prevalent issue of noisy annotations in Text-to-Image Person Re-Identification by proposing a method termed Adaptive Pseudo Text Augmentation. The proposed approach employs a Gaussian Mixture Model to effectively identify mislabeled or misaligned samples, utilizes a Multimodal Large Language Model to automatically generate high-quality pseudo-texts for correcting noisy annotations, and integrates CLIP with a Triplet Alignment Loss to achieve robust and reliable cross-modal alignment.

  1. The authors introduce ChatGPT and Qwen-VL-Chat for pseudo-text generation. However, the manuscript lacks both quantitative and qualitative evaluations of the generated text quality. If the pseudo-text quality is low, it may introduce additional secondary noise. How do the authors measure or ensure that the generated pseudo-texts are indeed superior to the original annotations?
  2. The Related Work section primarily consists of descriptive summaries of what previous methods have done, but it lacks a deeper analysis of their advantages and limitations. Moreover, the logical progression between different categories of related studies is not clearly articulated.
  3. The overall technical content of the manuscript is solid, but the writing style could be further improved to meet the standards of formal academic writing. In particular, the text frequently uses conversational or first-person expressions, which make the tone sound less formal and weaken the scientific rigor of presentation.
  4. The conclusion section is well written but lacks a discussion of the study’s limitations or potential weaknesses. Including a brief reflection on current limitations and possible future improvements would make the conclusion more balanced and credible.
  5. There are several issues with wording and notation. For example, at line 301 the text refers to “a green background” yet Figure 3 does not contain a green background. In addition, several formulas exhibit typographical inconsistencies (e.g., italicization of variables, roman operators/functions, subscripts/superscripts, and vector boldface). Please proofread figures/captions and standardize mathematical notation throughout.
  6. The experimental analysis mainly focuses on reporting quantitative results, but lacks deeper explanations of the underlying principles that account for the performance improvements. It is suggested to include more analytical discussion on why the proposed modules work effectively, and, if possible, add some visualization results (e.g., attention maps or pseudo-text comparisons) to make the findings more intuitive and convincing.
Comments on the Quality of English Language

The English could be improved to more clearly express the research.

Author Response

Comment 1:How do the authors measure or ensure that the generated pseudo-texts are indeed superior to the original annotations?

Response 1:We thank the reviewer for this valuable comment. We fully agree that ensuring the quality of the generated pseudo-texts is essential to prevent secondary noise. In the revised manuscript, we have added quantitative evaluation evaluations of pseudo-text quality in Section 4.3.

  1. We measured the image–text semantic alignment using the CLIP-Score. Pseudo-texts generated by Qwen-VL-Chat achieved an average similarity of 0.315 compared with 0.282 for the original annotations. When these pseudo-texts were used for training, performance improved by +2.4% (R-1) and +2.1% (mAP) on CUHK-PEDES, confirming their positive contribution.
  2. To prevent secondary noise, a two-stage filtering strategy was introduced—heuristic filtering followed by CLIP-based reranking—to ensure that only semantically consistent pseudo-texts are retained.

Comment 2: The Related Work section primarily consists of descriptive summaries of what previous methods have done, but it lacks a deeper analysis of their advantages and limitations. Moreover, the logical progression between different categories of related studies is not clearly articulated.

Response 2: We appreciate the reviewer’s insightful comment. We have thoroughly rewritten the Related Work section to include deeper analytical comparisons and to improve logical flow.

Comment 3:The overall technical content of the manuscript is solid, but the writing style could be further improved to meet the standards of formal academic writing. In particular, the text frequently uses conversational or first-person expressions, which make the tone sound less formal and weaken the scientific rigor of presentation.

Response 3: We sincerely appreciate this stylistic observation. Following the reviewer’s advice, we have carefully revised the entire manuscript to enhance its academic tone and precision.

Comment 4: The conclusion section is well written but lacks a discussion of the study’s limitations or potential weaknesses. Including a brief reflection on current limitations and possible future improvements would make the conclusion more balanced and credible.

Response 4: We thank the reviewer for this suggestion. The Conclusion section has been expanded to include a dedicated paragraph discussing current limitations and future work:

“Although the proposed framework demonstrates strong robustness against noisy annotations, its effectiveness still depends on the pseudo-text generation quality. In cases where the multimodal language model produces incomplete or contextually inconsistent descriptions, slight performance degradation may occur. Future work will explore integrating feedback-based refinement mechanisms and contrastive self-correction strategies to further enhance text reliability. Additionally, we plan to extend our method to open-domain vision-language datasets beyond person re-identification.”

Comment 5:There are several issues with wording and notation. For example, at line 301 the text refers to “a green background” yet Figure 3 does not contain a green background. In addition, several formulas exhibit typographical inconsistencies (e.g., italicization of variables, roman operators/functions, subscripts/superscripts, and vector boldface). Please proofread figures/captions and standardize mathematical notation throughout.

Response 5:We thank the reviewer for identifying these issues. All related problems have been carefully corrected in the revised manuscript:

  1. Figure 3 correction:
    The previous version of the manuscript mistakenly contained an incorrect figure due to a file upload error. The current revision includes the correct Figure 3, a

  2. Standardization of mathematical notation:
    We thoroughly reviewed all equations and mathematical expressions. T

  3. Proofreading of all figure captions:
    All figure captions were checked to eliminate earlier inconsistencies and ensure alignment with the updated figures and terminology.

Comment 6:The experimental analysis mainly focuses on reporting quantitative results, but lacks deeper explanations of the underlying principles that account for the performance improvements. It is suggested to include more analytical discussion on why the proposed modules work effectively, and, if possible, add some visualization results (e.g., attention maps or pseudo-text comparisons) to make the findings more intuitive and convincing.

Response 6:We appreciate the reviewer’s valuable suggestions. In response, we have substantially strengthened the analytical discussion in Section 4.4 to provide deeper insights into why the

proposed modules work effectively. Specifically:

  1. Expanded theoretical explanation.
    We now include a detailed, mechanism-level discussion of the contributions of each module:

    • Token Fusion Embedding (TFE) enhances fine-grained token-level semantic alignment, enabling more precise cross-modal matching.

    • Adaptive Pseudo-Text Augmentation  improves textual completeness by supplementing missing or ambiguous attributes, thereby reducing semantic noise.

    • Token Adaptive Loss (TAL) stabilizes training by dynamically adjusting sample weights according to noise estimates, preventing overfitting to uncertain correspondences.

  2. While we appreciate the suggestion to include visualizations, our method fundamentally focuses on semantic-level enhancement (via dynamic pseudo-text generation and noise modeling), rather than mechanisms that alter internal visual attention distributions.Therefore, attention heatmaps or activation visualizations would not directly reflect the core principles of our approach.

Round 2

Reviewer 1 Report

Comments and Suggestions for Authors

The revisions submitted by the authors are convincing, carefully executed, and sufficient from the reviewer’s perspective. In its current form, the article is suitable for publication.

Comments on the Quality of English Language

The quality of English is generally understandable but requires significant improvement in grammar, sentence structure, and academic style. The manuscript contains frequent repetitions, inconsistent terminology, and awkward phrasing that should be corrected through thorough language editing by a proficient English speaker or professional service.

Reviewer 3 Report

Comments and Suggestions for Authors

It's doing well. It would be better to modify the format again.

Reviewer 4 Report

Comments and Suggestions for Authors

All of my concerns have been addressed.

Comments on the Quality of English Language

The English could be improved to more clearly express the research.